# Calprotectin and Imbalances between Acute-Phase Mediators Are Associated with Critical Illness in COVID-19

**DOI:** 10.3390/ijms23094894

**Published:** 2022-04-28

**Authors:** Georgios Kassianidis, Athanasios Siampanos, Garyphalia Poulakou, George Adamis, Aggeliki Rapti, Haralampos Milionis, George N. Dalekos, Vasileios Petrakis, Styliani Sympardi, Symeon Metallidis, Zoi Alexiou, Theologia Gkavogianni, Evangelos J. Giamarellos-Bourboulis, Theoharis C. Theoharides

**Affiliations:** 1Intensive Care Unit, Korgialeneion-Benakeion Athens General Hospital, 115 26 Athens, Greece; georgekassianidis@gmail.com; 24th Department of Internal Medicine, ATTIKON University General Hospital, Medical School, National and Kapodistrian University of Athens, 1 Rimini Street, 124 62 Athens, Greece; thansiampanos@gmail.com (A.S.); gkavtheo@yahoo.gr (T.G.); 33rd Department of Internal Medicine, Medical School, National and Kapodistrian University of Athens, 115 27 Athens, Greece; gpoulakou@gmail.com; 41st Department of Internal Medicine, G. Gennimatas General Hospital of Athens, 115 27 Athens, Greece; geo.adamis@gmail.com; 52nd Department of Pulmonary Medicine, Sotiria General Hospital of Chest Diseases, 115 27 Athens, Greece; aggeliki.rapti@gmail.com; 61st Department of Internal Medicine, Medical School, University of Ioannina, 455 00 Ioannina, Greece; hmilioni@uoi.gr; 7Department of Medicine and Research Laboratory of Internal Medicine, National and European Expertise Center in Autoimmune Liver Diseases, General University Hospital of Larissa, 412 21 Larissa, Greece; georgedalekos@gmail.com; 82nd Department of Internal Medicine, Medical School, Democritus University of Thrace, 681 00 Alexandroupolis, Greece; vasilispetrakis1994@gmail.com; 91st Department of Internal Medicine, Thriasio General Hospital of Eleusis, 196 00 Magoula, Greece; lianasympa@hotmail.com; 101st Department of Internal Medicine, Medical School, Aristotle University of Thessaloniki, 546 21 Thessaloniki, Greece; metallidissimeon@yahoo.gr; 112nd Department of Internal Medicine, Thriasio General Hospital of Eleusis, 196 00 Magoula, Greece; z_alexiou@yahoo.gr; 12Laboratory of Molecular Immunopharmacology and Drug Discovery, Department of Immunology, Tufts University School of Medicine, Boston, MA 02111, USA; 13School of Graduate Biomedical Sciences, Tufts University School of Medicine, Boston, MA 02111, USA; 14Department of Internal Medicine, Tufts University School of Medicine and Tufts Medical Center, Boston, MA 02111, USA; 15Institute of Neuro-Immune Medicine, Nova Southeastern University, Clearwater, FL 33759, USA

**Keywords:** ARDS, COVID-19, cytokines, IL-18, IL33r, IL-38, S100A8/A9

## Abstract

The trajectory from moderate and severe COVID-19 into acute respiratory distress syndrome (ARDS) necessitating mechanical ventilation (MV) is a field of active research. We determined serum levels within 24 h of presentation of 20 different sets of mediators (calprotectin, pro- and anti-inflammatory cytokines, interferons) of patients with COVID-19 at different stages of severity (asymptomatic, moderate, severe and ARDS/MV). The primary endpoint was to define associations with critical illness, and the secondary endpoint was to identify the pathways associated with mortality. Results were validated in serial measurements of mediators among participants of the SAVE-MORE trial. Levels of the proinflammatory interleukin (IL)-8, IL-18, matrix metalloproteinase-9, platelet-derived growth factor (PDGF)-B and calprotectin (S100A8/A9) were significantly higher in patients with ARDS and MV. Levels of the anti-inflammatory IL-1ra and IL-33r were also increased; IL-38 was increased only in asymptomatic patients but significantly decreased in the more severe cases. Multivariate ordinal regression showed that pathways of IL-6, IL-33 and calprotectin were associated with significant probability for worse outcome. Calprotectin was serially increased from baseline among patients who progressed to ARDS and MV. Further research is needed to decipher the significance of these findings compared to other acute-phase reactants, such as C-reactive protein (CRP) or ferritin, for the prognosis and development of effective treatments.

## 1. Introduction

The SARS-CoV-2 coronavirus infects human cells by first binding to their surface receptor, angiotensin-converting enzyme 2 (ACE2), via its corona spike protein [1], leading to the development of COVID-19 [2,3]. This condition involves a complex immune response that includes the release of a “storm” of acute phase mediators [4,5,6]. This is a positive response of the host to prime recruitment of neutrophils and other inflammatory cells and associated production of cytokines and chemokines [7] but may reflect a worsening of the clinical situation if the cause is not eliminated. Prominent among them have been IL-6 [8,9,10] and IL-1β [11], but also the family of S100 alarmins, among which S100A8/A9 is otherwise known as calprotectin [12].

This knowledge led to early consideration of severe COVID-19 as a hyperinflammatory disorder and encouragement of the administration of modifiers of the biological response, such as dexamethasone, tocilizumab, anakinra and baricitinib, early in the course of management. Although this treatment strategy appears plausible, its rationale remains a field of active debate. This debate is stimulated by the contribution of acute-phase mediators to the immune defense and by the lack of knowledge as to whether hyperinflammation is still present at the time point of acute respiratory distress syndrome (ARDS) necessitating mechanical ventilation (MV). It is thus necessary to better identify what drives the transition from moderate and severe COVID-19 to severe ARDS and MV after the initial viral infection. This may be the only way to prevent or efficiently address the most ominous consequences of COVID-19.

In this paper, we investigated serum levels of alarmins, as well as pro- and anti-inflammatory molecules leading to ARDS and MV and how their presence tracks with disease severity.

## 2. Results

### 2.1. Study Participants

The sampling for the study took place between April and November 2020. Serum samples from 181 patients and 40 non-infected comparators were studied. Among the patient population, on the day of blood sampling, 19 patients were asymptomatic, 42 patients had moderate disease, 78 patients had severe disease, and 42 patients had ARDS and were on MV. Sampling took place within the first 24 h after admission. Patients with ARDS on MV had higher sequential organ failure assessment (SOFA) scores, lower respiratory ratios and higher circulating C-reactive protein and ferritin (Table 1). None of the 181 patients was in need of hemodialysis at the time of blood sampling. Patients with ARDS under MV were sedated; mean ± SD creatinine on the day of sampling was 1.0 ± 0.2 mg/dL; mean ± SD albumin was 3.2 ± 0.5 g/dL.

In the first stage of the study, the samples were analyzed in an effort to identify a set of mediators that may drive progression toward ARDS and MV. In the second stage of the study, the mediators were serially followed over time to define if the levels change over-time until ARDS and MV (Figure 1).

### 2.2. First Stage: Presence of Inflammation-Associated Mediators

Among the proinflammatory mediators, interleukin (IL)-1β, IL-6, IL-17, IL-33 and tumor necrosis factor alpha (TNFα) did not differ between the stages of severity. However, IL-8 and IL-18 levels were significantly higher among patients with critical ARDS necessitating MV compared to less severely ill patients. Among anti-inflammatory cytokines, IL-33r (soluble ST2) levels were significantly higher among patients with critical ARDS. IL-1ra (soluble receptor antagonist) levels were higher among patients with mild to critical COVID-19, although there was no significant difference between patients with severe and critical disease. IL-10 did not differ among patients and controls. IL-38 levels were decreased in asymptomatic patients relative to patients at more severe stages (Figure 2). Interferons did not differ between patients (Appendix A).

Platelet-derived growth factor (PDGF)-B and, to a lesser extent, PDGF-A were increased among critically ill ARDS patients; PAF did not differ. The same was the case for calprotectin and matrix metalloproteinase (MMP)-9. PAF and S100B did not differ among different stages of severity (Figure 3).

### 2.3. First Stage: Associations between Inflammatory Mediators and Progression into Acute Respiratory Distress Syndrome (ARDS) Necessitating Mechanical Ventilation (MV)

ROC curve analyses identified that IL-8, IL-18, MMP-9, IL-33r, PDGF-B and calprotectin were associated with ARDS necessitating MV (Figure 4A). The cut-offs of each of the six mediators providing the best trade-off of sensitivity and specificity were defined and entered into the equation of multivariate analysis. Only increased IL-33r, increased PDGF-B and increased calprotectin were found to be drivers of ARDS necessitating MV (Figure 4B).

### 2.4. First Stage: Associations between Biomarkers and 28-Day Mortality

Analysis involved hospitalized patients with moderate and severe COVID-19, as well as patients with ARDS necessitating MV. For this analysis, we decided to implement a pathway-like division of biomarkers into clusters. To this end, nine pathways were defined as follows: (a) IL-1ra activation, when both IL-1β and IL-1ra were above the median of the entire cohort; (b) IL-6 pathway, when IL-6 was above the median of the entire cohort; (c) IL-18 pathway, when IL-18 was above the cut-off defined for critical illness; (d) neutrophil activation, when at least two of the following conditions were met: IL-8 > 100 pg/mL, MMP-9 > 870 pg/mL or IL-17 above the lower limit of detection; (e) interferon pathway, when at least one of the three measured interferons was above the median of the entire cohort; (f) IL-33 pathway, when IL-33r was above 170 ng/mL; (g) IL-38 pathway, when IL-38 was above the median of the entire cohort; (h) PDGF-B pathway, when PDGF-B was above 2.7 ng/mL; and (i) calprotectin pathway, when calprotectin was above 7.8 μg/mL (Figure 5 and Figure 6A).

Multivariate ordinal regression analysis showed that the pathways of IL-6, IL-33 and calprotectin were associated with allocation into more severe states of the WHO-CPS after 28 days (Figure 6B). On the contrary, IL-1ra and IL-38 pathways were associated with allocation into less severe states of the WHO-CPS after 28 days. The IL-6, IL-33 and calprotectin pathways were independent drivers for mortality, acting synergistically. On the contrary, the IL-1ra and IL-38 pathways did not affect survival (Figure 6C,D).

### 2.5. Second Stage: Validation of the Role of S100A8/A9 (Calprotectin) in Progression into ARDS Necessitating MV

The SAVE-MORE trial enrolled patients who were in neither the ARDS nor MV category [13]. Among participants allocated to placebo treatment, samples collected serially from patients who progressed into ARDS and MV were analyzed. This was done because patient follow-up under placebo is considered to represent the progression of COVID-19 under the current standard-of-care management. At baseline, circulating levels of calprotectin and PDGF-B were similar among patients who eventually progressed into ARDS and MV and among those who did not progress into ARDS and MV (Figure 7A,B). On day 4, there was a trend toward increasing calprotectin from baseline among patients who developed ARDS and necessitated MV. This trend became a largely significant difference on day 7 (Figure 7C). No similar changes were found for PDGF-B (Figure 7D). Following ROC curve analysis, it was found that any increase in calprotectin from baseline was associated with the development of ARDS and MV. This association was proven to be independent of disease severity and of dexamethasone treatment after multivariate forward stepwise Cox regression analysis (Figure 7E). None of these patients needed hemodialysis before day 8 (Figure 7F).

## 3. Discussion

Our results highlight, for the first time, the conditions necessary for patients with COVID-19 to demonstrate the most worrisome features of critical illness with ARDS necessitating MV. The pathways of danger-associated molecular patterns (DAMPs), IL-33 (ST2) and coagulation drive ARDS and MV, whereas final outcome is dominated by excess of a proinflammatory state through the IL-6, IL-33 and DAMPs pathways, as well as by decreased levels of the anti-inflammatory pathways IL-38 and IL-1ra.

Several studies suggest that serum calprotectin is increased with COVID-19 severity. Twelve studies were subjected to systemic review [14] and eight studies were meta-analyzed [15]. Our findings regarding elevated serum levels of S100A8/A9 (calprotectin) are in line with previous results in the serum and in the bronchoalveolar lavage of COVID-19 patients [16,17], as well as with the results of the systemic review [14] and of the meta-analysis [15], showing, as we did, that calprotectin levels are higher among patients with the more severe state of disease. Most of these studies enrolled limited numbers of patients. Among these studies, only one failed to show that calprotectin is an indicator of unfavorable outcome. This study reported on the measurements of calprotectin of 222 hospital admissions in an emergency department; 25 patients had an unfavorable outcome, which was defined as a composite of need of non-invasive ventilation, MV or death [18]. Our study presents two main differences in design compared to previous publications: (a) the comparison of the trajectories of COVID-19 from asymptomatic to moderate severe and critical disease where calprotectin steadily increases; and (b) the independent association of calprotectin with the progression into critical illness and unfavorable outcome through a pathway-like approach, including multivariate analysis. Others have also recently reported that calprotectin is further increased from a state of non-invasive ventilation to the need of MV [19].

It is known that SARS-CoV-2 per se cannot elicit large amounts of inflammatory mediators leading to ARDS and MV [20]. However, rapid viral replication in the lungs leads to the destruction of lung epithelial cells and to the subsequent release of intracellular DAMPs, such as calprotectin. Our findings suggest that as the disease progresses and the patient worsens towards ARDS, increased calprotectin could contribute to the need for MV. This DAMPs pathway acts synergistically with the IL-6 and IL-33 pathways, mediating unfavorable outcomes. The crucial role of calprotectin in the pathogenesis of COVID-19 is further supported by a mouse model of lethal SARS-CoV-2 infection in which expression of *S100A8* in the lungs was increased, whereas infection by the influenza A virus, encephalomyelitis virus and herpes simplex virus were not accompanied by an increase in the expression of *S100A8*. Treatment of mice infected by SARS-CoV-2 with the calprotectin inhibitor paquinimod increased survival by 100% and attenuated infiltration of the lungs by neutrophils, suggesting a possible role of calprotectin for neutrophil chemotaxis and activation [21].

Although PDGF-B was not increased over time towards progression into ARDS and MV, the increased circulating PDGF-B described herein is interesting, given reports of widespread pulmonary microthromboses in lungs of deceased COVID-19 patients [22].

Our findings of increased levels of IL-1ra and IL-33r are intriguing, as serum levels of IL-1β and IL-33 levels were not significantly increased. This apparent contradiction may be explained by the autocrine mode of action of these cytokines following their production, implying that COVID-19 patients, especially those with severe disease, produce soluble IL-1ra and IL-33r (sST2) in an apparent effort to curtail the effect of IL-1β and IL-33, respectively. Elevated sST2 has also been reported in severe (especially non-surviving) patients with COVID-19 [23]. A recent publication reported that published scRNseq data from bronchoalveolar lavage fluid from patients with mild to severe COVID-19 contained a population of cells that produce IL-33 and correlate with severity of disease [24]. Two other publications reported a unique correlation of IL-12p70/IL-33 with disease severity [25] and increased expression of IL-33 in cultured epithelial cells infected with SARS-CoV-2 [26]. Taken together, these findings seem to imply that there may be local synthesis of IL-33, especially in the lungs, leading to reactive production of IL-33r (sST2). IL-33 can stimulate resident mast cells to produce impressive amounts of proinflammatory cytokines [27,28], and the findings of increased IL-33r should be interpreted as a counterbalance to the increased IL-33.

Similarly to the increased levels of IL-1ra and IL-33r, the anti-inflammatory IL-38 was increased only in asymptomatic patients as compared to non-infected comparators but decreased significantly in the more severe patient groups. This finding implies that patients cannot produce sufficient IL-38 to counteract the proinflammatory cytokines produced during COVID-19. IL-38 belongs to the IL-1 family [29] and has been reported to exhibit anti-inflammatory activity [30]. IL-38 exists intracellularly as a precursor in full-length form and must be cleaved at the N terminus before it is secreted extracellularly in an active form [31]. We showed that IL-38 can inhibit the release of IL-1β from cultured human microglia stimulated by bacterial lipopolysaccharide (LPS) [32].

The final outcome of patients appears to be determined by the excess activation of the IL-6, IL-33 and DAMP pathways, and the anti-inflammatory IL-1ra and IL-38 pathways are not sufficient to counterbalance them. A recent paper reported that elevated plasma levels of IL-6, IL-10 and IP-10 anticipated clinical progression of COVID-19 patients [33]. Our results are in line with the positive treatment outcomes with IL-6 receptor antagonists and with anakinra (the recombinant non-glycosylated form of IL-1ra), as they have shown the need to attenuate the increased IL-6 response and to increase the reduced IL-1ra responses [34]. Tocilizumab and sarilumab were shown to decrease the number of days under organ dysfunction and 28-day mortality in the REMAP-CAP [35] and RECOVERY [36] platform adaptive trials respectively. In these trials, the two IL-6 receptor antagonists were administered in patients already diagnosed with critical illness. In the open-label SAVE trial [37] and in the double-blind, randomized trial SAVE-MORE [13], anakinra was administered to hospitalized non-critical patients with plasma levels of the biomarker suPAR (soluble urokinase plasminogen activator receptor) 6 ng/mL or more; suPAR was used in these studies as a biomarker of an attack of the host by DAMPs.

Results of the large-scale RECOVERY study show that the use of dexamethasone improves outcomes in COVID-19 patients, mainly those requiring oxygen and under MV [38]. Part of the action of dexamethasone is associated with decreased neutrophil production of S100A8/A9 [39].

This is one of few studies reporting on coadministered drugs among mechanically ventilated patients with ARDS. Unfortunately, information on exact dosing and nutrition was not provisioned to be captured because such information was beyond the scope of the study. This can be conceived as a limitation.

## 4. Methods

### 4.1. Patient Cohorts

Studied patients were adults with molecular detection of SARS-CoV-2 and were classified into four groups using the WHO classification of COVID-19 severity: (a) asymptomatic, (b) hospitalized with moderate COVID-19, (c) hospitalized for severe COVID-19 or (d) hospitalized with acute respiratory distress syndrome (ARDS) necessitating mechanical ventilation (MV). Blood samples (10 mL) were collected in pyrogen-free tubes without anticoagulant (Becton Dickinson, Cockeysville Md) within the first 24 h after admission in the emergency department, the general ward or the intensive care unit. Patients had already participated as comparators, receiving usual care in the ACHIEVE [40], SAVE [37] and ESCAPE [41] studies. Patients were enrolled after written informed consent was provided by themselves or their legal representatives.

A volume of 10 ml of blood was also collected from comparators matched for age and sex with the patients and who were admitted to emergency departments for unspecified complaints. These comparators were adults without any specific diagnosis of disease; this was certified by a phone call after 30 days. Their inclusion was provisioned in the above studies, and they also signed a written informed consent.

In order to provide evidence for the role of the mediators for progression into ARDS necessitating MV, samples collected from 52 patients participating in the SAVE-MORE trial were studied. SAVE-MORE is a double-blind, randomized clinical trial where patients were allocated to treatment with placebo or anakinra plus standard of care to prevent progression into severe respiratory failure [13]. The studied samples were from patients with severe disease in need of treatment with oxygen and who were allocated to the placebo group so that comparative levels of the mediators at serial time points may dictate their role towards development into ARDS and MV. With this aim, only patients from the placebo group not under modulation with biologicals during all three time periods of sampling (baseline, day 4 and day 7) were studied; 19 patients progressed into ARDS and MV within the first 28 days; another 33 patients matched for age, gender and comorbidities who did not progress into ARDS and MV were studied as comparators.

### 4.2. Laboratory Assays

All blood samples were processed immediately, and the serum was stored at −80 °C until assayed. Levels of biomarkers were quantified by using commercially available kits of enzyme immunosorbent assays from Bio-Techne (R&D Systems, Minneapolis, MN, USA) according to the manufacturer’s instructions. The lower limits of detection were 16 pg/mL for tumor necrosis factor-alpha (TNFα), 8 pg/mL for interleukin (IL)-1β, 31 pg/mL for IL-1ra (receptor antagonist), 40 pg/mL for IL-6, 31 pg/mL for IL-8 and IL-10, 8 pg/mL for IL-17, 62 pg/mL for IL-18, 31 pg/mL for IL-33r (receptor), 62 pg/mL for IL-38, 156 pg/mL for interferon (IFN) alpha; 78 pg/mL for IFNβ, 156 pg/mL for IFNγ, 80 pg/mL for platelet activation factor (PAF), 313 p/mL for platelet-derived growth factor (PDGF)-A and PDGF-B, 61 pg/mL for S100A8/A9, 31 pg/mL for matrix metallopeptidase (MMP)-9 and 46 pg/mL for S100B.

### 4.3. Study Endpoints

The primary study endpoint was to define a set of cytokines and inflammatory mediators that are independently associated with progression into critical ARDS necessitating mechanical ventilation.

The secondary study endpoints were to define a set of cytokines and inflammatory mediators that are independently associated with the outcome of COVID-19 after 28 days, as this is expressed both by the strata of the WHO Clinical Progression Scale (CPS) and by mortality.

### 4.4. Statistical Analysis

Concentrations of biomarkers between groups of patients were compared using the Mann–Whitney non-parametric U test following Bonferroni correction for multiple testing. Receiver operator characteristic (ROC) curve analysis was performed for each of the measured mediators to define a cut-off that can significantly discriminate patients with ARDS necessitating MV; the areas under the curve of the ROC (AUC), 95% confidence intervals (CIs) and *p* values were determined. For significant variables, the cut-off concentration providing the best trade-off for sensitivity and specificity was determined by the Youden index. In order to define the variables that were drivers of ARDS necessitating MV, forward stepwise logistic regression analysis was performed. ARDS necessitating MV was entered into the equation as a dependent variable and all significant mediators at the predefined cut-offs as independent variables.

Mediators were divided with a pathway-like approach. Nine pathways were defined as follows: (a) IL-1ra activation using IL-1β and IL-1ra; (b) IL-6 pathway using IL-6; (c) IL-18 pathway using IL-18; (d) neutrophil activation using IL-8, MMP-9 and IL-17; (e) interferon pathway using the three measured IFNs; (f) IL-33 pathway using IL-33 and IL-33r; (g) IL-38 pathway using IL-38; (h) platelet-activation pathway using PAF, PDGF-A and PDGF-B; and (i) calprotectin/alarmin pathway using S100A8/A9. The pathways were defined in a hierarchical order, using the cut-offs of ROC curve analysis as the first step in the case of significant mediators and the median levels of the entire cohort as the second step (Figure 5). Patients were subgrouped as belonging or not belonging to each of the nine pathways of activation (Yes/No). Comparisons of the WHO-CPS strata in each subgroup were performed by univariate ordinal regression analysis, followed by multivariate ordinal regression analysis. Next, 28-day mortality was compared between subgroups by Cox forward stepwise multiple regression analysis. In this analysis, 28-day mortality was the dependent variable, and variables significant for WHO-CPS were the independent variables.

The fold change of mediators determined to be associated with ARDS and MV on days 4 and 7 from baseline was calculated and compared by the Mann–Whitney U test. The cut-off of fold change that was associated with ARDS and MV was defined by the area under the ROC curve using the Youden index. The value of this cut-off was further validated by forward stepwise Cox regression analysis. Progression into ARDS and MV was the dependent variable; the defined cut-offs, baseline severity and dexamethasone treatment were the independent variables.

## 5. Conclusions

Multivariate ordinal regression showed that IL-6, IL-33 and calprotectin pathways were significantly associated with worse outcome. Further research is needed to decipher the significance of these findings compared to other acute-phase reactants, such as C-reactive protein (CRP) or ferritin. Our results further suggest that the innate increase in IL-1ra and IL-33r, along with the drop in IL-38, is not sufficient to halt the progression of COVID-19. Attenuating the IL-6, IL-33 and DAMP pathways while administering IL-1ra, IL-33r and, potentially, recombinant IL-38, seems to be the most promising strategy to combat severe COVID-19.

## Figures and Tables

**Figure 1 ijms-23-04894-f001:**
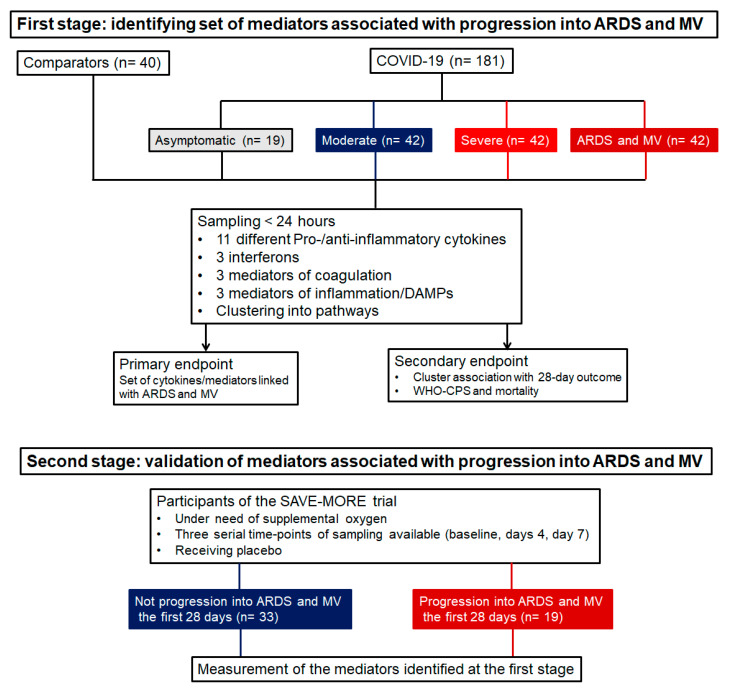
Study flow chart. Abbreviations: ARDS = acute respiratory distress syndrome; MV = mechanical ventilation; *n* = number of patients.

**Figure 2 ijms-23-04894-f002:**
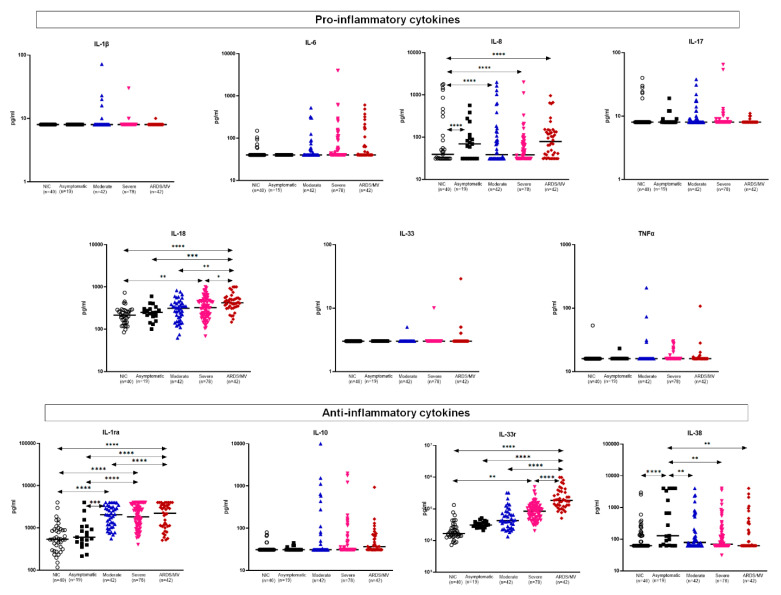
Concentrations of acute-phase mediators. Mediators are divided into proinflammatory and anti-inflammatory cytokines. Dot plots with horizontal lines indicate the median of each group. The number of subjects evaluated is listed in parentheses. Double arrowhead lines indicate comparisons between groups. Only statistically significant differences are indicated as follows: * *p* < 0.05; ** *p* < 0.01; *** *p* < 0.001; **** *p* < 0.0001. Abbreviations: ARDS = acute respiratory distress syndrome; IL = interleukin; MV = mechanical ventilation; *n* = number of patients; NIC = non-infected comparators; TNFα = tumor necrosis factor alpha.

**Figure 3 ijms-23-04894-f003:**
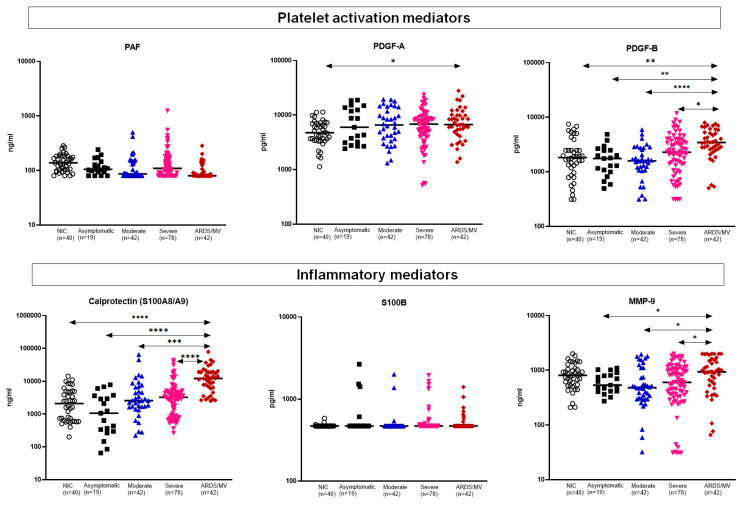
Concentrations of platelet-associated mediators and of other measured inflammatory mediators. Dot plots with horizontal lines indicate the median of each group. The number of subjects evaluated is listed in parentheses. Double arrowhead lines indicate comparisons between groups. Only statistically significant differences are indicated as follows: * *p* < 0.05; ** *p* < 0.01; *** *p* < 0.001; **** *p* < 0.0001. Abbreviations: ARDS = acute respiratory distress syndrome; MMP: matrix metalloproteinase; MV = mechanical ventilation; *n* = number of patients; NIC = non-infected comparators; PAF = platelet activation factor; PDG = platelet-derived growth factor.

**Figure 4 ijms-23-04894-f004:**
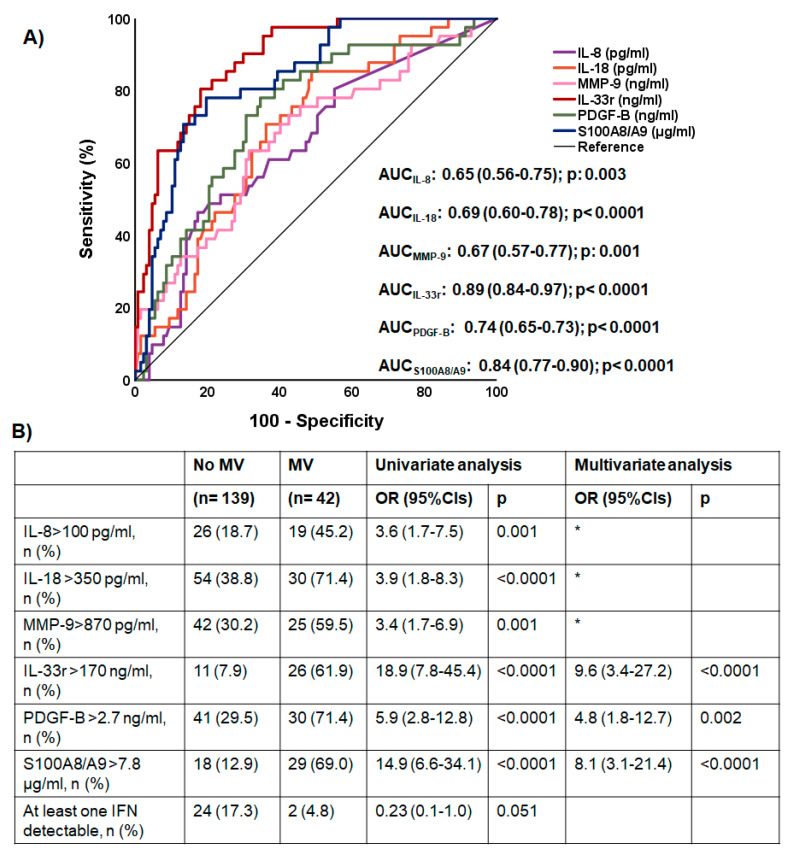
Main acute-phase mediators associated with acute respiratory distress syndrome (ARDS) necessitating mechanical ventilation. (**A**) Receiver operator characteristic (ROC) curves of the six studied mediators with a statistically significant area under the ROC curve (AUC) for the detection of ARDS necessitating mechanical ventilation (MV). The AUCs, their 95% confidence intervals and significant *p* values are provided. The ROC curves of the other 14 mediators measured are not provided because they were not statistically significant. (**B**) Following Youden index analysis, the cut-offs of each of the six mediators shown in panel A were determined. These cut-offs were used in univariate and multivariate forward stepwise logistic regression analyses to select the mediators associated with ARDS necessitating MV. * variables excluded after three steps of forward analysis. Abbreviations: CI = confidence interval; OR = odds ratio.

**Figure 5 ijms-23-04894-f005:**
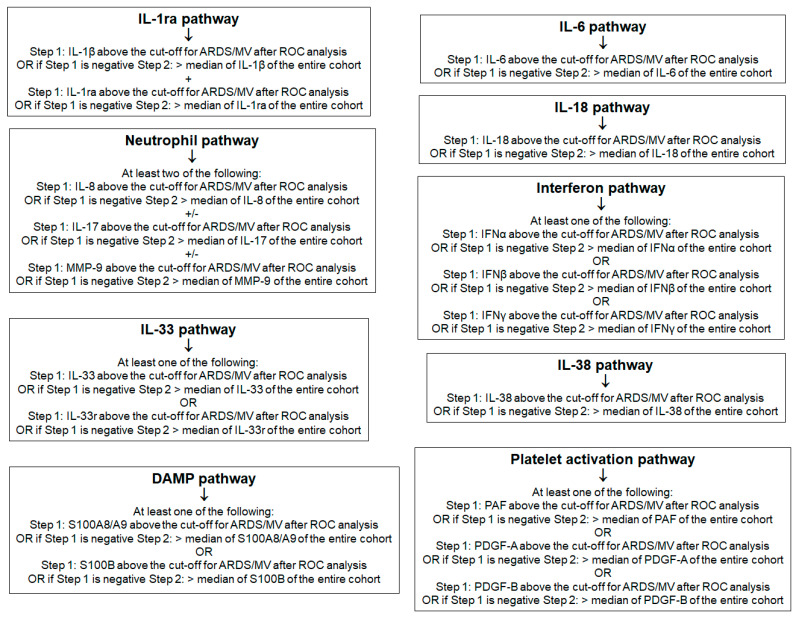
Principles of pathway analysis. Abbreviations: ARDS = acute respiratory distress syndrome; DAMP = danger-associated molecular pattern; IFN = interferon; IL = interleukin; MV = mechanical ventilation; PAF = platelet activation factor; PDG = platelet-derived growth factor.

**Figure 6 ijms-23-04894-f006:**
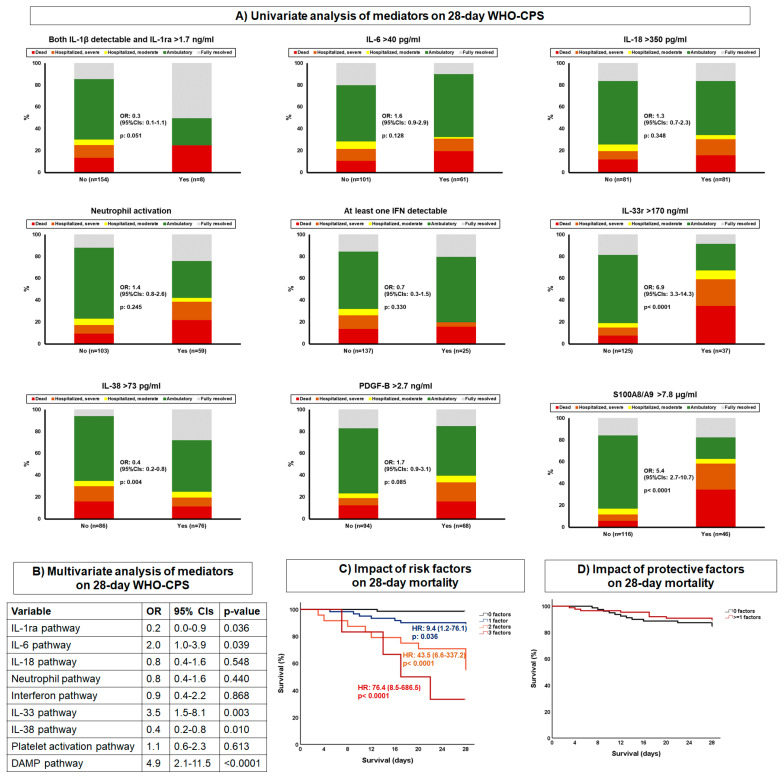
Main associations between mediators and 28-day outcome. The analysis involved patients with COVID-19 hospitalized with moderate COVID-19 and severe COVID-19, as well as those hospitalized with ARDS necessitating mechanical ventilation. (**A**) Univariate ordinal regression analysis of mediators associated with patterns of activation or inhibition of the inflammatory response. Ordinal regression analysis was performed. The principles for the selection of variables used to classify pathways are provided in Figure 5. (**B**) Multivariate ordinal regression analysis of pathways of the inflammatory response. (**C**) The three mediators associated with high risk for worse outcome according to the multivariate ordinal regression analysis were analyzed by Cox regression for their impact on 28-day mortality. (**D**) The two mediators associated with low risk for worse outcome according to the multivariate ordinal regression analysis were analyzed by Cox regression for their impact on 28-day mortality. Abbreviations: ARDS = acute respiratory distress syndrome; CI = confidence interval; IL = interleukin; *n* = number of patients; OR = odds ratio; PDGF = platelet-derived growth factor.

**Figure 7 ijms-23-04894-f007:**
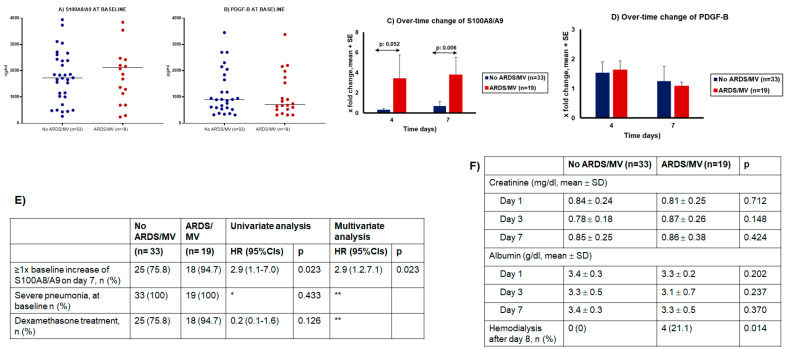
S100A8/A9 (calprotectin) as an independent variable of progression into acute respiratory distress syndrome (ARDS) necessitating mechanical ventilation (MV). The analysis involved serial measurement among 52 patients with severe COVID-19 participating in the SAVE-MORE study. (**A**) Circulating concentrations of calprotectin at baseline. No significant differences were found between patients who progressed into ARDS in need of MV and those who did not progress into ARDS in need of MV. (**B**) Circulating concentrations of PDGF-B at baseline. No significant differences were found between patients who progressed into ARDS in need of MV and those who did not progress into ARDS in need of MV. (**C**) Changes of circulating concentrations of calprotectin on days 4 and 7 of follow-up from baseline. The *p* values of comparisons are provided. (**D**) Changes of circulating concentrations of PDGF-B on days 4 and 7 of follow-up from baseline. No significant differences were found between patients who progressed into ARDS in need of MV and those who did not progress into ARDS in need of MV. (**E**) Univariate and multivariate forward stepwise Cox regression analysis of variables associated with progression into ARDS and MV; * HR cannot be calculated because one value is zero; ** excluded after two steps of forward analysis. (**F**) Serum concentrations of creatinine and albumin. The *p* values of comparisons are provided. Abbreviations: CI = confidence interval; HR: hazard ratio; PDGF = platelet-derived growth factor.

**Table 1 ijms-23-04894-t001:** Demographic characteristics of participants in the first stage of the study.

	Comparators	Asymptomatic	Moderate	Severe	ARDS and MV
Number	40	19	42	78	42
Age (years) mean ± SD	58.3 ± 15.8	59.5 ± 10.8	55.3 ± 14.8	61.4 ± 13.9	64.9 ± 12.8
Male gender, *n* (%)	28 (70)	12 (63.2)	25 (59.5)	56 (71.8)	35 (83.3)
SOFA, mean ± SD	NA	NA	0.9 ± 0.9	2.3 ± 1.6	6.5 ± 2.5
CCI, mean ± SD	1.0 ± 0.2	1.3 ± 1.9	2.3 ± 2.3	2.6 ± 2.1	2.4 ± 1.5
Comorbidities, *n* (%)					
Type 2 diabetes mellitus	0 (0)	5 (26.3)	13 (31.0)	20 (25.6)	6 (14.3)
Chronic heart failure	0 (0)	0 (0)	3 (7.1)	5 (5.1)	0 (0)
Chronic renal disease	0 (0)	0 (0)	2 (4.8)	3 (3.8)	2 (4.8)
Coronary heart disease	0 (0)	0 (0)	5 (11.9)	8 (10.3)	6 (14.3)
Dyslipidemia	0 (0)	4 (21.1)	14 (33.3)	17 (21.8)	4 (9.5)
Hypothyroidism	0 (0)	2 (10.5)	7 (16.7)	7 (9.0)	3 (7.1)
Hypertension	0 (0)	2 (10.5)	16 (38.1)	24 (30.8)	7 (16.7)
Stroke	0 (0)	0 (0)	2 (4.8)	1 (1.3)	1 (2.4)
Atrial fibrillation	0 (0)	0 (0)	2 (4.8)	6 (7.7)	3 (7.1)
COPD	0 (0)	0 (0)	3 (7.1)	2 (2.6)	0 (0)
pO_2_/FiO_2_, mean ± SD	NA	NA	397.5 ± 84.3	274.5 ± 103.8	127.1 ± 72.2
White blood cell count, mean ± SD (/mm^3^)	NA	NA	5394.3 ± 2379.8	7000.1 ± 2945.8	11,977.8 ± 6094.2
CRP, median (IQR), mg/L	NA	NA	4.9 (30.6)	39.0 (80.1)	82.9 (173.5)
Ferritin, median (IQR), ng/mL	NA	NA	293.2 (534.2)	485.3 (832.5)	1189.5 (1686.8)
Administered drugs, *n* (%)					
β-lactamase inhibitor	NA	0 (0)	1 (2.4)	21 (29.2)	0 (0)
Ceftriaxone	NA	0 (0)	30 (71.4)	41 (56.9)	10 (23.8)
Ceftaroline	NA	0 (0)	0 (0)	17 (23.6)	10 (23.8)
Piperacillin/tazobactam	NA	0 (0)	8 (19.0)	10 (13.9)	16 (38.1)
Ceftaziidme/avibactam	NA	0 (0)	0 (0)	0 (0)	6 (14.3)
Glycopeptides/linezolid	NA	0 (0)	0 (0)	3 (4.2)	12 (28.6)
Remdesivir	NA	0 (0)	20 (47.6)	30 (41.7)	9 (21.4)
Dexamethasone	NA	0 (0)	4 (9.5)	72 (100)	42 (100)
Nor-adrenaline	NA	0 (0)	0 (0)	0 (0)	27 (64.3)
Furosemide	NA	0 (0)	0 (0)	0 (0)	29 (69.0)
Midazolam	NA	0 (0)	0 (0)	0 (0)	29 (69.0)
Fentanyl	NA	0 (0)	0 (0)	0 (0)	24 (57.1)
Propofol	NA	0 (0)	0 (0)	0 (0)	29 (69.0)
Dexmetomidine	NA	0 (0)	0 (0)	0 (0)	7 (16.7)
Cisatracurium	NA	0 (0)	0 (0)	0 (0)	7 (16.7)

Abbreviations: ARDS = acute respiratory distress syndrome; CCI = Charlson’s comorbidity index; CRP = C-reactive protein; COPD = chronic obstructive pulmonary disease; FiO_2_: fraction of inspired oxygen; IQR = interquartile range; MV = mechanical ventilation; NA = not available; pO_2_: partial oxygen pressure; SD = standard deviation; SOFA = sequential organ failure assessment score.

## Data Availability

Data are available from the corresponding author upon request.

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
