# Peer review of "Calprotectin and Imbalances between Acute-Phase Mediators Are Associated with Critical Illness in COVID-19"

_ijms, 2022, doi:10.3390/ijms23094894_

Round 1

Reviewer 1 Report

This is a reasonably well designed and well written study that describes serum levels of biomarkers (calprotectin, pro- and anti-inflammatory cytokines, interferons) expressed in patients with COVID-19 at different stages of severity (asymptomatic, moderate, severe and ARDS/MV).  The objective was to define the drivers to critical illness and the pathways associated with mortality. 

Minor criticisms:

  1. The manuscript doesn't fully cite the studies on association between calprotectin and severity of COVID-19. For example, there has been a meta-analysis describing 8 publications on this topic between 2020 and 2021. One of the studies by Mentzer et al (2020) did not support the value of calprotectin in predicting ICU requirement in COVID-19 patients. It may be valuable to compare the current findings against these other studies.
  2. There seems to be a suggestion in the literature that calprotectin acts distinctively during SARS-CoV-2 infection to drive chemotaxis of a
    group of aberrant neutrophils, resulting in cytokine storm. Can the authors comment on this in light of their findings?
  3. The authors should clarify whether the collection tubes contain heparin, EDTA, or citrate.

Author Response

REVIEWER 1

  • This is a reasonably well designed and well written study that describes serum levels of biomarkers (calprotectin, pro- and anti-inflammatory cytokines, interferons) expressed in patients with COVID-19 at different stages of severity (asymptomatic, moderate, severe and ARDS/MV).  The objective was to define the drivers to critical illness and the pathways associated with mortality. 

Reply: We would like to thank this reviewer for his favorable comments.

  • The manuscript doesn't fully cite the studies on association between calprotectin and severity of COVID-19. For example, there has been a meta-analysis describing 8 publications on this topic between 2020 and 2021. One of the studies by Mentzer et al (2020) did not support the value of calprotectin in predicting ICU requirement in COVID-19 patients. It may be valuable to compare the current findings against these other studies.

Reply: One big part of the Discussion has been fully re-written in order to address your comment and to add the suggested reference. The manuscript now reads on pg21, ln1-18: “Several studies are suggesting that serum calprotectin is increased with COVID-19 severity. Twelve studies are subject to systemic review [14] and eight studies are meta-analyzed [15]. Our findings of elevated serum level S100A8/A9 (calprotectin) are in line with previous results in the serum and in the bronchoalveolar lavage of COVID-19 patients [16, 17] but also of the results of the systemic review [14] and of the meta-analysis [15] showing, as we did, that calprotectin levels are greater among patients with the more severe state. Most of these studies are enrolling limited number of patients. Among these studies, only one has failed to show that calprotectin is an indicator of unfavorable outcome. This study reported on the measurements of calprotectin of 222 hospital admissions at the emergency department; 25 patients had unfavorable outcome defined as a composite of need of non-invasive ventilation, MV or death [18]. Our study is presenting two main differences in design compared to previous publications: a) the comparison of the trajectories of COVID-19 from asymptomatic, to moderate, severe and critical disease where calprotectin is steadily increasing; and b) the independent association of calprotectin with the progression into critical illness and unfavorable outcome through a pathway-like approach including multivariate analysis. Others have also recently reported that calprotectin is further increased from a state of non-invasive ventilation to the need of MV [19].” New added references are references 14, 15, 18 and 19. Reference 15 is the suggested meta-analysis and reference 18 the reference by Mentzer et al. Consecutive enumeration of the remaining references has been changed accordingly.

  • There seems to be a suggestion in the literature that calprotectin acts distinctively during SARS-CoV-2 infection to drive chemotaxis of a
    group of aberrant neutrophils, resulting in cytokine storm. Can the authors comment on this in light of their findings?

Reply: The manuscript now reads on pg22, ln1-8: “The crucial role of calprotectin in the pathogenesis of COVID-19 is further supported by a mouse model of lethal SARS-CoV-2 infection. The expression of S100A8 in the lung was increased; infection by the influenza A virus, encephalomyelitis virus and herpes simplex virus was not accompanied by increase of the expression of S100A8. Treatment of mice infected by SARS-CoV-2 with the calprotectin inhibitor paquinimod increased survival by 100% but also attenuated infiltration of the lung by neutrophils showing a role of calprotectin for neutrophil chemotaxis and activation [21].”. A new reference, reference 21 has been introduced. Consecutive enumeration of the remaining references has been changed accordingly.

  • The authors should clarify whether the collection tubes contain heparin, EDTA, or citrate.

Reply: Measurements were done in serum. This is clarified on pg, ln of the revised manuscript.

Reviewer 2 Report

This is a retrospective analysis of blood samples collected from SARS-CoV-2 infected patients  with difeerent clinical severity following infection up to ARDS and mechanical ventilation (MV).

Several serum proteins,cytokines and chemokines were measured.

Serum obtained from 40 patients without SARS-CoV-2 infection and from 52 patients belonged to      the placebo group from a trial (SAVE-More trial) testing the effectivity of anakinra to inhibit interleukin 1beta in patients with severe disease in need of treatment with oxygen.In this group of placebo treated patients blood sampling was performed at base line,at day 4 and at day 7.Interestingly serum level of acute-phase mediators like IL-1beta,IL-6 and TNF alpha did not differ between the different stages of severity at base line.Serum level oft he chemokine Interleukin-8 and of IL-18 were higher in patients with critical ARDS with MV.

Il-10 did not differ among patients and controls as it did serum level of interferons.The level oft he chemokine IL-8 and calprotectin correlates with ARDS and MV and IL-33r,PDGF-beta and calprotectin were found tob e driver of ARDS and MV.

Multivariate ordinal regression analysis showed that the pathways of IL-&,of IL-33 and calprotectin were associated with the more severe states oft he WHO-CPS after 28 days and the pathways of IL_&,IL-33 and calprotectin were independen drivers for mortality acting sinergistically.The SAVE-MORE placebo tretment patients were followed up as they were considered to represent progression of SARS-Cov-2-infection under the current standard of care current management.

In these atients at day 4 a trend to an increase of calprotectin serum level from base line in the patients who progressed to ARDS and MV.This trend became significant at day 7.This association was proven to be independent of disease severity

Comment

1.a)The title cites the proinflammatory and antiinflammatory biomarkers.It has to be realized that the definition „proinflammatory“ for IL-6,IL1 and TNF alpha is not appropriate as those cytokines belong tot he main acute-phase mediators a spart of the first defense line in cases of tissue injury.

  b)….determine clinical illness in COVID-19.Tis part of the title suggests a link of causality which is not

     justified by the data presented.

2.to the abstract a)The first sentence of the abstract is not complete.It is infact very well known that hypoalbuminemia is the best negative prognostic marker independently of age and comorbidities.

b)determination of serum level of different proteins does not help to establish any causal relationship with clinical developement towards ARDS and need of mechanical ventilation.

c).calpotectin is an acute-phase protein very much alike CRP.Serum level of CRP increased in the 3 clinical severity groups as it did ferritin,which is also a positive acute -phase protein.

d).the meaning oft he word „drivers“ is not clear

e)in the conclusion of the abstract calprotectin could be replaced by CRP or ferritin which is not new and has not helped to develop new treatments.

3.To introduction.

  1. a) SARS-CoV-2 coronavirus infects human cells ….involves a complex immune response that includes the release of a „storm“ of pro-inflmmatory cytokines associated with poor prognosis..The acute-phase response is not a sign of bad prognosis and those cytokines are not „pro-inflammatory“ (see above and Malik I, Biology 2022).

b)it has to be admitted that use of immunosuppressant to avoid release of the acute-phase cytokines a spart of the first libne of defence aiming to eliminate the aggressor to to minimize tissue damage  was and still is not justified.

c)..the need to identify  the real drivers…. The first driver is the virus.It is questionable if the virus continues to remain the driver of progression of the clinical picture toward ARDS and mechanical ventilation

4.To Methods:a)the data from the treatment group oft he trial SAVE-More should also be used.

   b)the standard of care treatment fort he different clinical situations including those patients under MV should be specified .The description should include all the drugs,the fluid and nutrition administration.

   c)Albumin and creatinin serum levels should be reported especially for the patients of the trial.

   d)the report should also mention dialysis treatment.

In summary

a)the manuscript should be reshaped (from title to discussion) considering the literature on acute-phase reaction.

b) the necessary very important clinical and laboratory informations should be given.

Author Response

REVIEWER 2

  • This is a retrospective analysis of blood samples collected from SARS-CoV-2 infected patients with different clinical severity following infection up to ARDS and mechanical ventilation (MV). Several serum proteins, cytokines and chemokines were measured. Serum obtained from 40 patients without SARS-CoV-2 infection and from 52 patients belonged to the placebo group from a trial (SAVE-More trial) testing the effectivity of anakinra to inhibit interleukin 1beta in patients with severe disease in need of treatment with oxygen. In this group of placebo treated patients blood sampling was performed at baseline, at day 4 and at day 7. Interestingly serum level of acute-phase mediators like IL-1beta, IL-6 and TNF alpha did not differ between the different stages of severity at baseline. Serum level of the chemokine Interleukin-8 and of IL-18 were higher in patients with critical ARDS with MV. Il-10 did not differ among patients and controls as it did serum level of interferons. The level of the chemokine IL-8 and calprotectin correlates with ARDS and MV and IL-33r, PDGF-beta and calprotectin were found to be driver of ARDS and MV. Multivariate ordinal regression analysis showed that the pathways of IL-8, of IL-33 and calprotectin were associated with the more severe states of the WHO-CPS after 28 days and the pathways of IL8, IL-33 and calprotectin were independent drivers for mortality acting sinergistically. The SAVE-MORE placebo treatment patients were followed up as they were considered to represent progression of SARS-Cov-2-infection under the current standard of care current management. In these patients at day 4 a trend to an increase of calprotectin serum level from base line in the patients who progressed to ARDS and MV. This trend became significant at day 7. This association was proven to be independent of disease severity. The title cites the proinflammatory and antiinflammatory biomarkers. It has to be realized that the definition „proinflammatory“ for IL-6,IL1 and TNF alpha is not appropriate as those cytokines belong to the main acute-phase mediators a spart of the first defense line in cases of tissue injury.

Reply: We wish to thank this reviewer for all his insightful comments. As you may see, we have thoroughly revised our manuscript following his suggestions.

  • b)….determine clinical illness in COVID-19.Tis part of the title suggests a link of causality which is not justified by the data presented.

Reply: We thank you for your comment. The title has now been changed into “Calprotectin and imbalances between pro- and anti-inflammatory biomarkers are associated with critical illness in COVID-19”.

  • to the abstract a)The first sentence of the abstract is not complete. It is in fact very well known that hypoalbuminemia is the best negative prognostic marker independently of age and comorbidities.

Reply: We have changed this sentence according to your suggestions.

  • determination of serum level of different proteins does not help to establish any causal relationship with clinical development towards ARDS and need of mechanical ventilation.

Reply: We thank you for your comment. We have now edited our manuscript thoroughly to outline that our findings point towards associations and not causal relationships.

  • calpotectin is an acute-phase protein very much alike CRP. Serum level of CRP increased in the 3 clinical severity groups as it did ferritin, which is also a positive acute -phase protein.

Reply: We thank you for your comment. As also stated above, we have now edited our manuscript thoroughly to outline that our findings point towards associations and not causal relationships.

  • the meaning of the word „drivers“ is not clear

Reply: This has been changed into “associations” in both the abstract and the manuscript.

  • in the conclusion of the abstract calprotectin could be replaced by CRP or ferritin which is not new and has not helped to develop new treatments

Reply: We thank you for your comments. The last sentence of the abstract has changed into “Further research is needed to decipher if these findings provide new information compared to the other acute phase reactants for the prognosis and development of effective treatments.”

  • To introduction a) SARS-CoV-2 coronavirus infects human cells ….involves a complex immune response that includes the release of a „storm“ of pro-inflammatory cytokines associated with poor prognosis. The acute-phase response is not a sign of bad prognosis and those cytokines are not „pro-inflammatory“ (see above and Malik I, Biology 2022).

Reply: This is now addressed on pg4, ln6-8 of the revised manuscript. The suggested reference has been introduced and cited as reference 7. Consecutive enumeration of the remaining references in the manuscript has been changed accordingly.

  • b) it has to be admitted that use of immunosuppressant to avoid release of the acute-phase cytokines as part of the first line of defence aiming to eliminate the aggressor to minimize tissue damage was and still is not justified.

Reply: The manuscript now reads on pg4, ln14-19: “Although this treatment strategy appears promising, its rationale is a field of active debate. Th debate is stimulated by the major contribution of the pro-inflammatory reaction to the immune defense and by the lack of evidence if when the patients progresses into acute respiratory distress syndrome (ARDS) necessitating mechanical ventilation (MV) hyper-inflammation remains or not.”

  • c)..the need to identify  the real drivers…. The first driver is the virus. It is questionable if the virus continues to remain the driver of progression of the clinical picture toward ARDS and mechanical ventilation

Reply: This has been rephrased on pg4, ln20 of the Introduction to better reflect what has been suggested by this reviewer.

  • To Methods: a)the data from the treatment group of the trial SAVE-More should also be used.

Reply: We cannot include all these data since they make part of one already published manuscript. The information on albumin and creatinine that you have asked us is provided on pg18, ln7-9 of the revised manuscript.

  • the standard of care treatment for the different clinical situations including those patients under MV should be specified .The description should include all the drugs, the fluid and nutrition administration.

Reply: This information has been added in the revised Table 1.

  • Albumin and creatinine serum levels should be reported especially for the patients of the trial. The report should also mention dialysis treatment.

Reply: This information is now provided on pg18, ln7-9 of the revised manuscript.

  • In summary a)the manuscript should be reshaped (from title to discussion) considering the literature on acute-phase reaction; b) the necessary very important clinical and laboratory information should be given.

Reply: All the suggested modifications were done. Furthermore, the Discussion was enrihed

Round 2

Reviewer 2 Report

Authors introduced some changes tu satisfy the criticisms of this reviewer.

Further and more important changes and informations are needed which preclude publication.

a)Title:the term pro-inflammatory is mesleading,it should be replaced by acute-phase mediators

b)Abstract:calprotectin,similar to CRP and Feritin was serailly increased..

c)introduction line 60 pro-inflammatory sghould be replaced by acute-phase

lines 61,62:this is a positve response of the host to prime recruitment of neutrohils and other inflammatory cells and production of cytokines and chemokines but may parallel a worsening of the clinical situation if the cause is not eliminated,

line 69:pro-inflammatory should be replaced....

d)Results Table 1 the table should contain the use of

vasopressors,diuretics,sedatives(narcotics) muscle relaxanx,pain killr(inclusing morphns) Nutrients(amount of calories and of proteins),fluids,cristalloids,albumin(?).

2.5 Second stage:values of creatinine and albumin at the different days should be given.The same is for hemodialysis.

Discussion,line 248:I would suggest to say "a possible role of calrpotectin.. (as only one publiction is available).

The role of corticosteroids should also be discussed as these drugs inhibit acute-phase response...should be discussed

Round 3

Reviewer 2 Report

The authors made additional changes.

I still miss the additional drugs:diuretics,sedatives relaxans, vasopressors,

nutrients (especially for the ventilated patients.

Fig.2 stil contains the word "pro-inflammatory" which should be replaced by

acute-phase mediators

Author Response

  • The authors made additional changes. I still miss the additional drugs:diuretics,sedatives relaxans, vasopressors, nutrients (especially for the ventilated patients.

Reply: All information on drugs has now been added in the revised Table 1. Information on nutrition was not captured in our CRF.

  • 2 still contains the word "pro-inflammatory" which should be replaced by acute-phase mediators

Reply: We respectfully believe that in this Figure this change should not been done in the Figure but in the legend. The legend now reads that measurements of acute-phase mediators are reported and that they are divided into pro-inflammatory and anti-inflammatory cytokines. The reason is that anti-inflammatory cytokines are also acute phase reactants and not maintaining this distinction may lead the reader to confusion.

Round 4

Reviewer 2 Report

This is the first paper I know which describes the drugs  used during ICU and MV in patients primarely affected by SARS-CoV-2 infection.Please add the mean dosage of the single drugs (e.g furosemide) and the parenteral nutrition.

Author Response

  • This is the first paper I know which describes the drugs used during ICU and MV in patients primarily affected by SARS-CoV-2 infection. Please add the mean dosage of the single drugs (e.g furosemide) and the parenteral nutrition.

Reply: We do thank this reviewer for his comment. Unfortunately, this information is not captured in the CRF since it is beyond the goal of our study. This is now reported as a limitation on pg24, ln5-8 of the Discussion.